# Digital Communication Studies during the Pandemic: A Sociological Review Using Topic Modeling Strategy

**Alba Taboada-Villamarín** * and **Cristóbal Torres-Albero** *

Department of Sociology, Autonomous University of Madrid, 28049 Madrid, Spain
* Correspondence: alba.taboada@uam.es (A.T.-V.); cristobal.torres@uam.es (C.T.-A.)

**Abstract:** The health crisis triggered by COVID-19 has exerted a profound influence on both conventional communication methods and the manifestations of interaction within the virtual sphere. Gradually, studies on digital communication have taken on an increasingly prominent role in various social science disciplines that address determinants such as the crisis of misinformation or digital interaction in contemporary societies. This study aims to analyze the key research topics that sociology has addressed in relation to the pandemic, along with the level of innovation in the utilization of digital sources and analytical methodology. The analysis is grounded in the hypothesis that the effects of the pandemic have led the discipline of sociology to reassess and more fully integrate studies on digital communication. On this premise, a systematic review of studies sourced from the Web of Science (WoS) and Scopus databases was executed. Innovative computational methodologies were employed for the categorization of articles and the elucidation of principal research topics. Furthermore, this research scrutinized the principal digital platforms utilized in these investigations and assessed the extent of methodological innovation applied to data analysis. The outcomes unveiled a pronounced ascendancy in the prominence of communication studies during the pandemic. Nevertheless, it is noteworthy that the utilization of digital data sources in research remains surprisingly limited. This observation highlights a potential avenue for further exploration within the domain of sociological research, promising a more profound and contemporaneous comprehension of social phenomena amid times of crisis.

**Keywords:** COVID-19; literature review; machine learning



## 1. Introduction

The intersection of communication, politics, and opinion studies has progressively garnered the attention of the sociological gaze. The triad that synthesizes the universal concepts of "work", "communication", and "norm" (Lamo de Espinosa 2002) consolidates communication as the central axis of sociability, pointing to one of the main elements that constitute social reality and thus must be explained. In order to frame our analysis, it is essential to consider the role of communication throughout the history of societies, from the most primitive forms in which human language gave rise to the broad cultural diversity represented by various languages to the present digital social networks. In a schematic manner, and overlooking the complexity of such processes, we could consider an initial phase or "Rosetta Stone galaxy" that originates in the early moments of human grouping, marked by the crucial milestone of the invention of writing in its various formats and the alphabet. This phase extends to its materialization in the form of books (Vallejo 2019).

Subsequently, we move toward the period or "Gutenberg galaxy" characterized by the significance of the invention of movable-type printing which forms the basis for the widespread dissemination of written communication. This leads to a universal trend toward literacy, the emergence of newspapers, and ultimately, the formation of public opinion alongside the establishment of civil and political rights.

The evolution continues with the phase or "Marconi galaxy" introducing the significant novelty of the relevance of modern science and technology, expressed in the first wave of information and communication technologies (telegraph, wired telephony, radio, and television). This phase is characterized by the resulting centrality of mass media. Finally, we arrive at what could be termed the "Internet galaxy" (Castells 2001), encompassing an array of contemporary technoscientific applications.

Within this latest phase, the second generation of information and communication technologies stands out, manifested in the initial triad formed by the personal computer, the Internet, and mobile telephony, as well as their more recent hybrids (smartphones and tablets). We observe here, as social processes resulting from this technoscientific impact, the emergence of digital social networks (Torres-Albero 2022).

This final stage constitutes the undeniable challenge that engages the social sciences, representing a new dimension that has not yet been fully explored within existing theoretical frameworks. In this context, researchers (Raab 2022; Geise and Waldherr 2021) underline the urgent need to delve deeper into digital communication studies from a sociological perspective, emphasizing the importance of a multidisciplinary approach in addressing this field of study.

Digital mediation in communication processes encounters certain obstacles that recent research has categorized as methodological and material limitations, attributable to a lack of knowledge in managing and accessing information. This information, presented in an unstructured manner in the virtual environment, demands advanced technology for its proper processing. On the other hand, the traditional focus on demoscopic orientation in sociology, involving techniques like surveys or opinion polls for gathering quantitative data, has progressively given way to cultural case studies. These emphasize the need to understand the dynamics of needs and power structures influencing specific population groups (Schulz et al. 2023).

The pandemic has significantly accentuated challenges associated with information and communication in the digital realm from a sociological perspective (Kidd et al. 2023). This phenomenon is clearly evidenced by a noteworthy surge in information consumption, which has become a universalized occurrence in the Western world following the spread of COVID-19 (Van Aelst et al. 2021). This study aims to adopt a broader perspective to analyze the transformative impact of the health crisis on political and social communication in virtual environments. It conducts a thorough examination of the main research themes that have concerned sociologists regarding the pandemic through a systematic literature review that emphasizes the study of the application of new methodologies in this field of research. It seeks to verify the hypothesis that COVID-19, by highlighting emerging forms of digitally mediated communication, has underscored the importance of studying these spaces from a sociological discipline, whether due to restricted access to other media during the pandemic or due to the growing significance that they have acquired in offline life and in traditional political settings. It is relevant to note that this research has been conducted through a global comparison of topics addressed by sociology in relation to COVID-19, with the aim of determining the quantitative relevance of studies on digital communication to various fields of study.

## 2. Communication during the Pandemic

The health crisis triggered by the COVID-19 pandemic has had profound impacts on all dimensions of society globally. In the context of the modern world, there were no precedents for such extensive mobility restrictions and social isolation practices. These events not only sparked deep sociopolitical debates but also involved significant cultural aspects, establishing themselves within complex reflective logics about essentially civic and ethical matters. In some cases, new barriers to access to healthcare resources emerged, or existing ones were exacerbated (Núñez et al. 2021; Pujolar et al. 2022). In various nations, on a global scale, there has been extensive discourse regarding the applicability of laws crafted to restrict the free flow of commerce in favor of public security policies, as well

as civil-military collaborations and health strategies. In this context, the role of states has been pivotal in terms of communication and the dissemination of information to contain movement flows and consumption activities (Gibson-Fall 2021). These issues provoked a strong reaction from liberal far-right factions, focusing their argument on freedom of movement and action, particularly concerning public spaces (Macip and Yuguero 2022).

During the extended months of the state of alarm in various geographical locations, it became essential to transform the communication, work, and cohabitation practices of social groups to maintain some normalcy in society. The distinction between different population groups based on professional profile, academic status, generational cohort, and health status became the main determinant of differences among citizens. This led to the creation of distinct and defining normative logics among social groups. The need for governments in democratic scenarios to establish complex strategies which rest on the principle of the universal equality of all human beings (Lewkowicz et al. 2022) resulted in the implementation of broad and sophisticated monitoring and control mechanisms, where communication, dialogue, and information dissemination played a key role.

Among the varied studies conducted regarding the diverse communicative and trust-building strategies implemented by different governments worldwide (Radwan and Mousa 2020; Arcila-Calderón et al. 2021; Kim and Kreps 2020; Kim 2023; Sanders 2020; Vardavas et al. 2021), it is possible to identify four fundamental imperatives within the scope of pandemic management:

1. Informing the population about the SARS-CoV-2 virus and disseminating health and, in some cases, scientific information, as it was essential for the population to understand the risks and dangers of an unknown virus (Salido Cortés and Massó 2021).
2. Communicating the measures and variety of regulations applicable to different sectors of the population in terms of adapting to the new normal, directly impacting families' daily lives.
3. Legitimizing and helping the population understand the necessity and social justice underpinning these measures.
4. Providing guidelines for the population to continue, as far as possible, with their work activities, particularly regarding educational study plans, health plans, and consumption.

From this perspective, communication and information were the common denominators in the prevention and containment strategies of the health crisis, deployed by central institutions holding the officiality of information and the trust of the civilian population. However, beyond the range of dialogue between corporations and governments, social sciences placed special emphasis on opinion meters and the recording of demoscopic studies on the feelings and degree of agreement of the population, as well as the internal and external communication dynamics developed during the pandemic.

Empirical research focused on observing social interactions mediated by the digital environment, such as changes in consumption through digital businesses (Giordani and Rullani 2020), different educational dynamics (Strielkowski 2020), and the formation of "default digital families" (Hantrais et al. 2021), among others. These findings contributed to the notion of the pandemic as a conjunctural point that has driven a "digital revolution" in society. The discourse on the health crisis and its effects on interaction and communication highlighted the imperative need to rethink existing social narratives, underscoring a significant change in the structure and dynamics of society both before and after the pandemic. Theoretically, this change was characterized by a greater dependence on digital technologies in various aspects of daily life, leading to transformations in the way people interact, work, learn, and relate online.

Although comparative studies on communication in this natural experiment have been of great importance, the confirmation of the frameworks in which the pandemic was managed, considering the structures of information and communication technologies, categorically refutes this thesis (Torres-Albero 2022). This is because, without understanding

that this revolution was already consolidated at the time the pandemic appeared, it would be impossible to understand the way it has developed.

The data indicating that the pandemic marked a turning point in general digital communications do not confirm the widespread and consolidated adoption of new technologies in everyday life. Instead, they reveal that these skills were already widely integrated into people's daily practices. Since it became impossible to continue life in a mixed format between digital and physical presence, the online digital space became the central element in human interactions. This was manifested, for example, in the increased time spent on digital platforms or the more frequent use of video calls, online shopping, or virtual physical activity tracking (Saud et al. 2020; Fernández-Rovira and Giraldo-Luque 2021; Parker et al. 2021). However, this did not imply an increase in the number of users of these technologies, nor their transformation into the main communication channel, as this phenomenon was already present before the pandemic, as shown by multiple data sets (EGM-AIMC 2023).

In general terms, the population spent more time connected in virtual environments, but it remains to be seen whether these levels of use of digital platforms are sustainable and, ultimately, if this is a direct product of the pandemic. On the contrary, it should be noted that this turning point is determined by other equally significant issues in the context of the information society from a social science perspective.

This research starts from the hypothesis that, focusing on sociological studies, although it can probably be extended to other social disciplines, studies on digital communication and information became much more relevant during the pandemic. This could be due, firstly, to the difficulty of studying emerging social phenomena outside the virtual space, considering that these phenomena in the virtual space mostly focus on communication and opinion, and secondly, to the need to measure opinion in spaces where they are truly influential and usually translate into changes in political agendas, such as what was being debated at that time about COVID-19, lockdowns, travel, etc., being discussed, for example, on social networks like Twitter.

There have been significant cases, for instance, where the effects of social media have had a major impact on Western democracies, ranging from the spread of fake news to the proliferation of conspiracy theories, culminating in historical landmarks such as the assault on the Capitol in the United States (Gounari 2022).

The confirmation of this theory would imply, in addition to the information it provides, the establishment of an epistemic consequence for the social sciences themselves. This is not limited to asserting that the pandemic has brought about a 'digital revolution' in societies, but rather, it has drawn the attention of disciplines such as sociology to the need to study the social mechanisms present in virtual environments. Moreover, it has highlighted the importance of applying and consolidating techniques for the extraction and analysis of information that are still lagging in the social sciences compared to other disciplines and the private sphere regarding studies of opinion and large-scale social behaviors.

## 3. Materials and Method

In order to elucidate the formulated hypothesis, this study has set two primary objectives: (1) to identify the main research topics in the sociological field regarding the health crisis originating by COVID-19, and (2) to assess to what extent these investigations have been oriented toward aspects related to 'digital' studies. This study examines the extent to which they utilized digital platform data sources along with methodological innovations and computational approaches to achieve their goals.

To this end, a systematic review of the existing literature has been chosen, resorting to emerging approaches to conduct massive literature analyses through computational techniques. These approaches have proven to be highly successful in understanding the current state of affairs in topics with a high density of publications, allowing for the identification of trends and patterns in research (some examples can be consulted in Karami et al. 2020; Cao et al. 2023; Taboada Villamarín 2024).

This approach involves the application of two sophisticated natural language processing algorithms: the K-means clustering algorithm and the Latent Dirichlet Allocation model. This section meticulously details the procedure used for the collection of scientific publications, which serve as secondary data. Subsequently, the process of data cleaning and analysis is described.

Furthermore, in the following pages, we strictly adhere to the guidelines established by authors Müller-Hansen et al. (2020). These guidelines are aimed at ensuring transparency, replicability, reproducibility, rigor, and validation in the implementation of codes and machine learning techniques in social research.

### 3.1. Data Source and Document Selection Process

To conduct a comprehensive extraction of all publications made since the onset of the pandemic in the field of sociology related to COVID-19 (see Table 1), an analysis was undertaken using the Journal Citation Reports database (JCR) focusing on all journals indexed in the specialization of sociology. This process resulted in the identification of 209 journals. This initial criterion specifically allowed us to delineate those publications belonging to the field of sociology, ensuring the inclusion of journals that also have interdisciplinary intersections with other fields of study.

**Table 1.** Literature review: search period, sources, search criteria, and subsample.

| Search Period | 2020–2022 (3 Years) |
| --- | --- |
| Sources | The following databases were used:<br>1. Web of Science: https://www.webofscience.com (accessed on 1 December 2022)<br>2. Scopus: https://scopus.com/ (accessed on 1 December 2022).<br>Criteria for journals indexed in sociology:<br>JCR: https://jcr.clarivate.com/jcr/browse-journals?app=jcr&referrer=target%3Dhttps:%2F%2Fjcr.clarivate.com%2Fjcr%2Fbrowse-journals&Init=Yes&authCode=null&SrcApp=IC2LS (accessed on 1 December 2022) |
| Search terms and syntax | All documents found in WoS and Scopus that met the condition that the word "covid" appears either in the title, in the abstract or in the keywords were extracted (both those mentioned by the authors and those referenced by the magazine that are collected in WoS and Scopus).<br>Search syntax in WoS:<br>TS = COVID.<br>Search syntax in SCOPUS:<br>TITLE-ABS-KEY (COVID). |
| Type of documents | Articles, books, book chapters, communications to conferences, editorials, notes, letters, and other materials. |

Source: authors.

Subsequently, the Web of Science (WoS) and Scopus databases were consulted, owing to their reputation for compiling high-impact publications on an international level. An exhaustive search was conducted with the requirement that the term "COVID" appear in the title, abstract, or keywords, both as proposed by the authors and as assigned by the journals. This search process yielded a total of 413,289 documents in WoS and 374,987 in Scopus, covering the period from January 2020 to December 2022.

In a subsequent phase, a rigorous selection of documents was carried out, limiting the scope exclusively to publications within the field of sociology. This operation resulted in the identification of a total of 1677 documents in the Web of Science (WoS) database and 2167 documents in Scopus. Afterwards, a meticulous process of cleaning and preparation of both databases was undertaken for their integration, culminating in the creation of a unified database upon which the relevant analyses will be conducted. To determine the

final number of documents included in our study universe, all publications with duplicate DOIs were eliminated, and a thorough review of authors, titles, and publication journals was conducted to identify potential duplicate articles without DOIs. As a result of this process, a total of 2200 documents were counted as an integral part of our study. The document selection procedure is detailed in Figure 1.

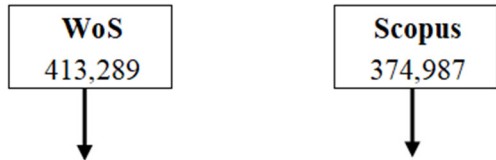

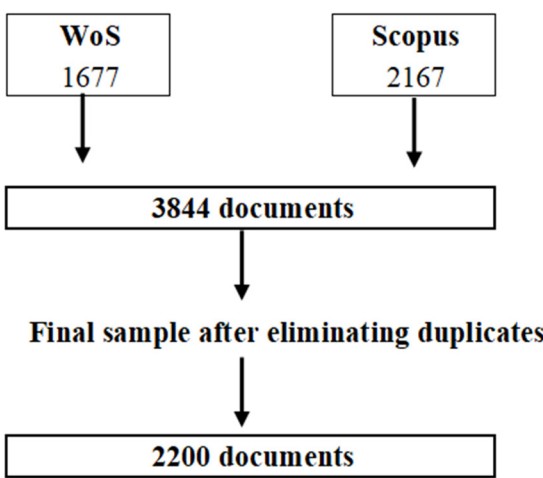

**Figure 1.** PRISMA flow diagram.

*3.2. Text Cleaning*

The topic modeling process involved using the title and abstract of each document as the textual database. This content was consolidated into an individual textual corpus for each document, thereby facilitating a more effective thematic analysis. A crucial stage in natural language processing (NLP) is the proper cleaning of the text. This process involves removing non-essential elements such as punctuation marks and words without significant analytical value, such as prepositions and articles, commonly identified in the literature as 'stopwords' (Sarica and Luo 2021).

For the removal of stopwords and the performance of tokenization (see Section 3.3), the natural language toolkit (NLTK) library was used. Additionally, the word "covid" was specifically excluded from the corpus, given its use in the study's search criteria. The most frequent terms and phrases identified in the document set are presented in Figures 2 and 3, providing a quantitative view of lexical prevalence in the analyzed corpus.

Following the initial review of the most frequently used words in the documents, the top 50 were manually analyzed, understanding their context and assessing which combinations do not provide real value to the comprehension of the information present in the publications. In this manner, combinations such as "All right reserved" or "Limited, trading as Taylor & Francis Group" were eliminated, which are commonly found at the end of abstracts and do not contribute relevant information to the subject of study. The words or combinations of words removed can be consulted in Table 2.

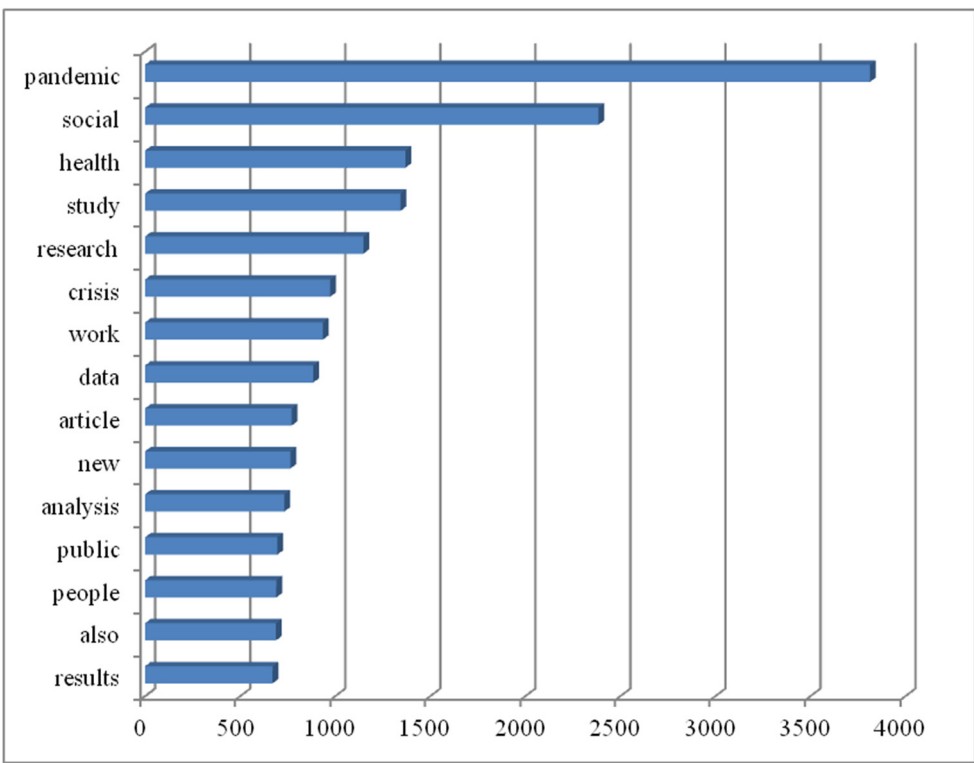

**Figure 2.** Most frequently used words in the entire set of documents before cleaning.

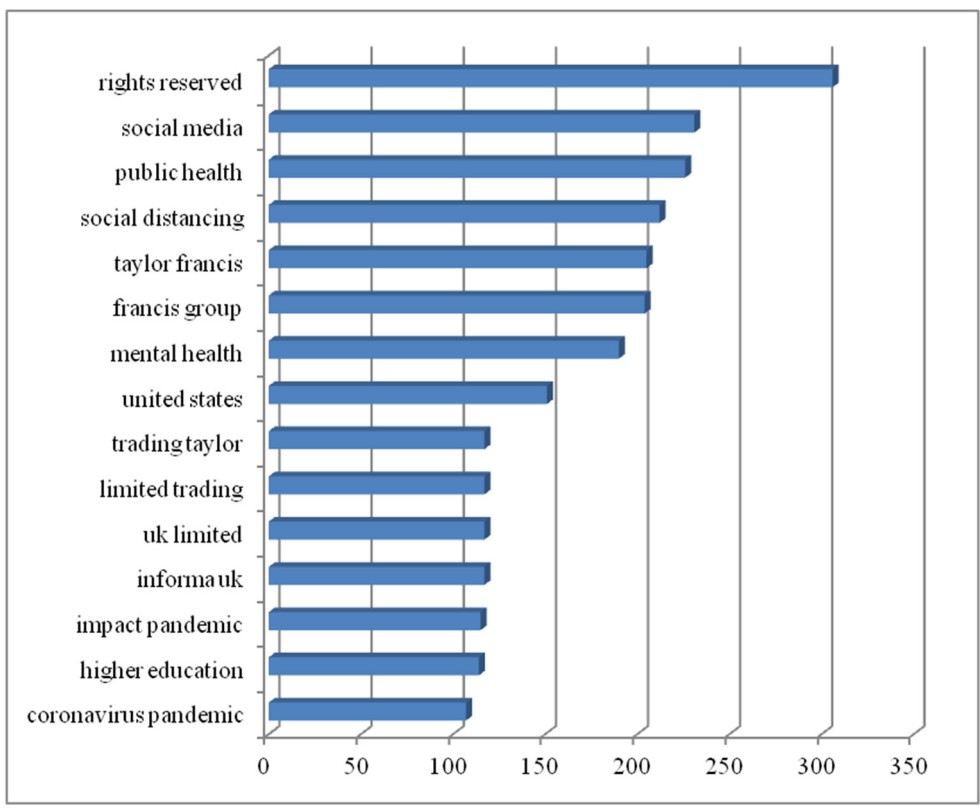

**Figure 3.** Most frequent word combinations in the entire set of documents before cleaning.

**Table 2.** Words or word combinations removed.

| |
|---|
| All right reserved |
| Taylor Francis |
| Francis Group |
| Informa uk |
| UK limited |
| Limited trading |
| Trading taylor |
| Research |
| Article |
| Analysis |
| Also |
| Results |
| Paper |
| Based |
| Findings |
| Author |
| Author published |
| University press |
| Group llc |
| Springer nature |
| Elsevier Ltd. |
| Authors licensee |

Source: authors.

### 3.3. Data Analysis

In order to understand the main topics investigated by sociology in relation to COVID-19, the following sections provide details on data preparation and the application of algorithms for topic modeling in our text corpus. Different strategies were evaluated in each of the following steps with the aim of being compared and validated.

### 3.3.1. Tokenization

In natural language processing (NLP), tokenization is the preliminary step to converting words into numeric vectors, enabling algorithms to work with them. It involves breaking the text into units (Saleem et al. 2021). For instance, the phrase "Online learning and emergency remote teaching" is tokenized into the following individual words: "online", "learning", "emergency", "remote", and "teaching", with each word becoming a token. However, there are multiple ways to perform this division, such as dividing the text into n-grams, where "n" represents the number of words and "gram" refers to the token itself. According to the literature consulted (Bird et al. 2009), tokenization using bi-grams often yields good results when applying algorithms like LDA. Using the previous example, bi-gram tokenization would appear as follows: "online learning", "learning and", "and emergency", "emergency remote", and "remote teaching".

For our analysis, we experimented with both types of tokenization, using bi-grams and uni-grams, allowing us to observe the impact of compound words and their context on inferring the topic.

### 3.3.2. Clustering Using K-Means

After the text corpus was tokenized and vectorized, our approach involved a preliminary step before topic modeling. Following the evidence of improved results, we applied the K-means algorithm (Gualda Caballero et al. 2023). This unsupervised classification algorithm allows data to be grouped into different clusters based on their proximity, resulting in greater consistency in the prominent words highlighted by the topic modeling. Since K-means does not rely on prior labels, we had to determine the number of clusters into which the documents should be grouped, evaluating this number using the "Elbow" method (James et al. 2013).

### 3.3.3. Implementation of Topic Modeling

The Latent Dirichlet Allocation (LDA) algorithm is the most popular choice for topic modeling (Jelodar et al. 2019). For its application, we decided to compare two libraries, Gensim and Scikit-learn (Sklearn), both available as statistical packages in Python. These two libraries work similarly, although we found differences in computation time, methods for visualizing results, and slight variations in the output words, with no significant impact on the final results.

Continuing with the classical methodology of thematic analysis in social sciences, we emphasized the interpretation of prominent words by manually reviewing sets of articles to verify the reliability of the results and to infer the topics. Additionally, comparing the models provided greater confidence in the results, as they consistently yielded similar outcomes, reinforcing the final labeling of topics.

### 3.3.4. Digital Communication Studies

Objective 2 aimed to evaluate the extent to which the analyzed academic research focused on aspects related to digital communication studies. In this context, articles were required to incorporate the analysis of a digitally mediated phenomenon, exploring its impacts, transformations, or characteristics. The objective also sought to determine the extent to which these studies utilized digital platform data sources, along with innovative methodologies and computational approaches, to achieve their research goals.

To conduct this assessment, only documents classified as "research articles" were selected, excluding reviews, research notes, and other document types, resulting in a sample of 1500 articles. Subsequently, articles containing the keywords "online", "digital", "remote", and their variants, in both the title and abstract, were filtered. This process generated a sample of 456 articles which underwent a thorough manual analysis to identify the research objective, methodology, and data sources. In cases where abstracts lacked essential information, a full article reading was conducted.

The information gathered during the manual review facilitated the creation of a Python program dictionary, encompassing all identified digital platforms, including instant messaging applications, e-commerce platforms, online services, etc. This dictionary was utilized to filter the overall database and label each document according to the platform and data source used, with detailed results provided in Section 4.2.

## 4. Findings

The current study aimed to understand the quantitative significance of studies on communication and media for the discipline of sociology, particularly in the context of researching the COVID-19 pandemic, compared to other subjects of study. This research analyzed a total of 2289 documents, finding that 78% of the journals, categorized as sociological or related to the field of sociology according to the JCR index, covered the topic. This implies that 22% of high-impact journals had not published any work mentioning COVID-19 even two years after the onset of the health crisis.

Regarding the timeline of publications, the original query included the year 2019; however, as expected, the first publications did not appear until 2020. The bulk of the research was published in the following two years, mainly in 2021. In 2022, while a similar number of articles were produced as in the previous year, this number slightly declined.

### 4.1. Main Research Topics of the COVID-19 Pandemic

The primary objective set forth in this study was to identify the main topics investigated in the sociological discipline regarding the health crisis caused by COVID-19. Through the use of natural language processing (NLP) techniques and the application of the textual analysis algorithms mentioned in Section 3, we were able to identify six major groups of general themes established as follows: 1. information and mass media, 2. health, 3. family, work, and care, 4. education, 5. tourism and economic effect, and finally, 6. food crisis and territorial systems.

As can be seen in Figure 4, the K-means algorithm provides a representation of how the documents are grouped according to affinity in the themes. Some themes show greater transversality across all clusters, as a manual evaluation of the publications reveals texts that can be easily classified into more than one cluster. However, this approach seems to respond well to the general themes that have been investigated in the sociological discipline regarding the health emergency. Table 3 presents the words and word combinations highlighted by the models that have led us to the inference and labeling of the main themes. Additionally, the table includes the number of documents and the percentage relative to the total that are grouped in each of the clusters. The two major themes that stand out in the research, 1. "Information and Media" and 2. "Health", account for 40.8% and 33.5%, respectively, of the analyzed works and are further disaggregated into 1.1 political polarization, racial segregation, and religion, 1.2 sports and economy, and 1.3 fake news and social media. Regarding the health cluster, it is divided into the subclusters 2.1 mental health and medical care, 2.2 public policies, vaccines, and trust, and 2.3 network analysis. Below, we develop the main lines addressed in each of these themes.

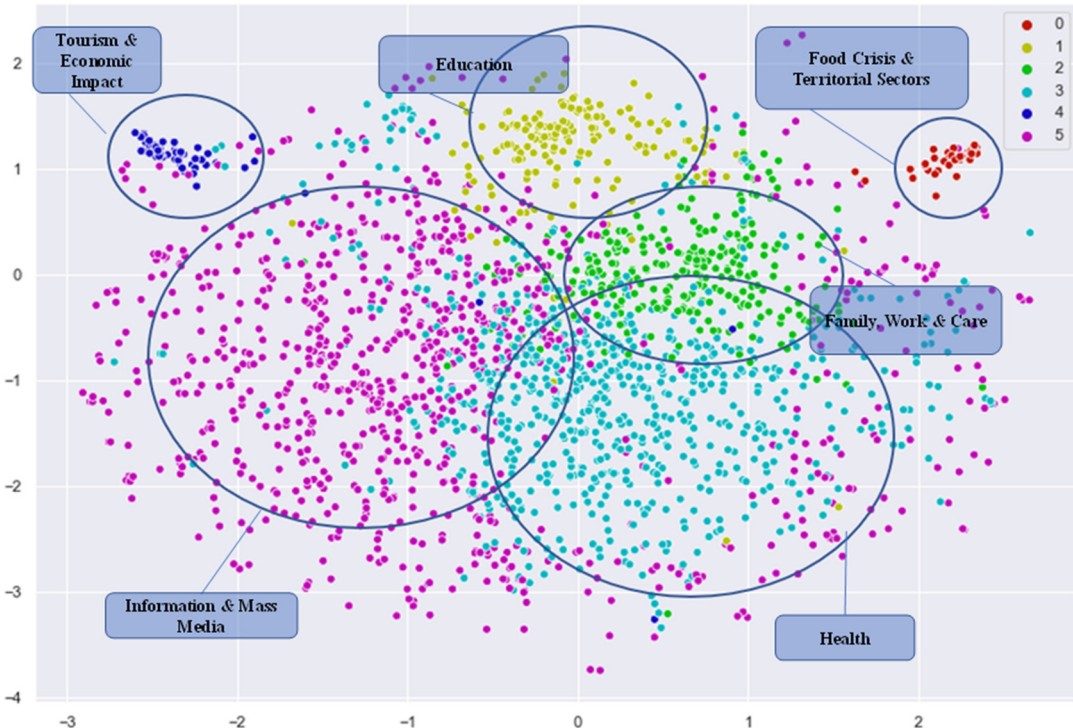

**Figure 4.** Scatter plot of the analyzed documents following the classification provided by K-means.

**Table 3.** Classification and analysis of topics.

| Topics | Number of Articles | % | Keywords |
|---|---|---|---|
| **1. Information & Mass Media**<br>1.1. Political polarization, racial segregation and religion<br>1.2. Sports and economy<br>1.3. Fake News and Social Networks | 898 | 40.8 | social, crisis, research, new, media, pandemic, problems, article, age, perspectives, online, digital, social media, study, analysis, critical, risk, market, hate, racial, asian, queer, black, change, climate, global, social, crisis |
| **2. Health**<br>2.1. Mental health and medical care<br>2.2. Public policies, vaccines, and trust<br>2.3. Network Analysis | 736 | 33.5 | health, study, pandemic, data, social, mental health, trust, patients, dicriminations, medical, risk, news, indigenous, science, social science, vaccines, public, perceived |

**Table 3.** *Cont.*

| Topics | Number of Articles | % | Keywords |
|---|---|---|---|
| **3. Family, Work & Care** | 274 | 12.5 | care, family, women, health, children, pandemic, discrimination, transnational, home, labor, gender, black, work, occupations, mobility, black women, parents, bitrh, labor, childbirth, school, stress |
| **4. Education** | 180 | 8.2 | online, education, teaching, teachers, students, learning, university, international, stress, young, physical, sources, digital, chilhood, programs |
| **5. Tourism and Economic Impact** | 69 | 3.1 | demand, development, local, pandemic, recovery, industry, domestic, impact, economic, post, tourism research, sustainable, destinations, stakeholders, regional, risk |
| **6. Food Crisis & Territorial Sectors** | 43 | 2.2 | systems, food insecurity, households, community, food systems, farmers, crisis, impacts, urban, global, supply, sustainable, poverty, opportunities |

Source: authors.

4.1.1. Information and Mass Media (40.8%)

Cluster 1, "Information and Mass Media", emerges as the most quantitatively significant in sociological research on COVID-19. Under this category, the studies are notable for addressing contemporary social issues prevalent in public debates generated on social media and news outlets. In this context, we identify three main research lines that converge in a significant portion of the analyzed studies.

Political Polarization, Racial Segregation, and Religion

The research within this group points to an increase in political polarization and a threat to Western democracies triggered by the socioeconomic crisis left in the wake of COVID-19. Studies focusing on racial segregation, the rise of the far right, and religion are prominent in this cluster. Many studies warn of the virtualization of religious rites and discourses and their connection with the American right, in contrast to the democratic line, which has advocated for caution regarding health aspects during the pandemic.

Sports and Economy

This cluster includes research related to the impact of the pandemic and confinement measures on social activities such as sports and leisure. The need to adapt social life to virtual spaces has led to an increase in consumers of e-sports and other cultural and artistic products that played a significant role in alleviating the health crisis at its peak. Similarly, the implications that COVID-19 has unleashed on the economy for younger population cohorts lead to a transformation in how young adults and teenagers perceive their future possibilities in terms of career trajectories.

Fake News and Social Media

In the "Fake News and Social Media" cluster, we find research that collects data from platforms like Twitter, Facebook, and online newspapers. These studies analyze information flows during the pandemic, emphasizing the role played by conspiracy theories and fake news in the information dynamics of the civilian population as they stay informed about the health emergency. This cluster shares political analyses about polarization and the rise of the far-right with cluster 1.1.

4.1.2. Health (33.5%)

The health cluster is established as the second major theme addressed by researchers on COVID-19 in the field of sociology. Although issues related to health and medical care are transversal to most publications due to the nature of the subject of study, this cluster

gathers studies that concretely address health problems, highlighting dimensions of 1. mental health and healthcare system, 2. public policies, vaccines, and trust systems, and 3. network analysis.

Mental Health and Healthcare

This subtheme carries the most quantitative weight in the health cluster. Multidisciplinary works are found that study the medical and psychosocial consequences of the pandemic, especially those aspects derived from confinement, anxiety in the face of a global emergency, and uncertainty toward the "new normality". Particularly noteworthy are studies on mental health pointing to an increase in emotional instability and anxiety. Collaborations from the field of sociology in the study of medical pathologies caused by the COVID-19 virus are also found.

Public Policies, Vaccines, and Trust

The publications in this subcluster focus, on the one hand, on the study of public policies carried out by governments and local authorities in relation to the management of the health crisis and, on the other hand, on health technologies such as vaccines. In both approaches, research is found that explores the trust of the civilian population in political and health institutions in the context of the pandemic.

Network Analysis

This last dimension includes works on the classic analysis of social networks, analyzing the ways the virus spreads, especially through consumers in the international market, medical collaboration networks, and interpersonal interaction networks.

### 4.1.3. Family, Work, and Care (12.5%)

Documents grouped under the "family, work, and care" cluster address as a central theme the problems in reconciling telework forced by the health crisis with care in the family and domestic environment. In this line, research highlights the gender gap in being able to balance paid employment and care work, particularly in families with children. Additionally, attention is paid to the severe problems of stress and mental health that workers may suffer in this attempt to balance parenting and work life, the problems derived from school closures for reconciliation, or the invisibility of domestic workers and elderly care exacerbated by the crisis. Further, research on social structure and labor market addressing gender and racial inequalities is found, observing that jobs with greater risk and insecurity in relation to COVID-19 are occupied by migrants and/or black people.

### 4.1.4. Education (8.2%)

In the "Education" cluster, the main challenges faced by the educational system in adapting classes and learning in a virtual environment during school closures caused by confinement are found to be addressed. On the one hand, studies analyzing experimental pedagogical methods through online resources, as well as the study of new virtual tools and platforms used by teachers in seeking new ways of communication with students, are found. Secondly, structural determinants in access to these resources and the follow-up of teaching by families are addressed, pointing out inequalities and difficulties in less resourceful cores. These investigations cover all educational levels, from kindergarten to higher university levels.

### 4.1.5. Tourism and Economic Impact (3.1%)

The works collected in this cluster center their research on the impact that the health crisis has had on the tourist market and its main economic effects. The research addresses the transformation of the industry, including studies on tourist behavior, economies dependent on the sector, experiences of local sellers, and mobility and transport policies. Many investigations delve into the costs and benefits of the pandemic, as well as the viability of

new, more environmentally and locally friendly forms of tourism. Some works advocate the critical questioning of tourism models and the possible perverse effects that have been accentuated during the crisis, highlighting this sector as one of the most affected and re-evaluating possible more respectful future alternatives for the involved parties.

4.1.6. Food Crisis and Territorial Systems (2.2%)

The "Food Crisis and Territorial Systems" cluster group focuses on the sociology of food, the agri-food industry, and territorial dichotomies between urban and rural environments. Part of the research points to changes in eating habits during the pandemic, especially focusing on inequalities due to economic and racial reasons, difficulty in accessing food, and different levels of information about health and nutrition. On the other hand, these investigations also highlight the traditional crisis suffered by the agricultural industry in the global market, which was even more unprotected in the context of the pandemic. New ways of and alternatives in food production that address the difficulties faced by small and medium-sized farms to compete in the international industry during and after the health crisis are proposed.

*4.2. Digital Communication Studies*

The second purpose of our study aimed to assess the extent to which the analyzed academic research has focused on aspects related to digital communication studies. Furthermore, the objective was to determine the degree to which these studies employed digital platform data sources, as well as methodological innovations and computational approaches, to achieve their goals. In order to conduct this analysis, only documents classified as "research articles" were exclusively selected, filtering out reviews and research notes, among others. The analyzed sample comprised 1500 articles.

Within this subset, 30.4% had the primary research goal of analyzing issues that emerged during the pandemic, with a focus on a digital dimension. For instance, in clusters such as education or mental health, there is a notable presence of articles investigating the transfer of daily actions, interpersonal communications, and the learning of sports methods exclusively through the online environment. These studies explore the new features of this mediation and its impact on health or the quality of work.

From a quantitative perspective, the cluster compiling the highest number of studies related to this subject is labeled as "1. Information & Mass Media", with 164 documents. Secondarily, we find the "2. Health" cluster with 137 documents, followed by the "4. Education" cluster with 95 documents, the "3. Family, Work, & Care" cluster with 53 documents, and in a residual manner, the "6. Food Crisis & Territorial Sectors" cluster and the "5. Tourism & Economic Impact" cluster with 6 and 2 documents, respectively.

However, out of this set of studies, only 172 documents (11.47% out of n = 1500) utilized digital data, i.e., derived from digital platforms such as social networks, blogs, websites, etc., to fulfill their objectives. Additionally, 73 studies collected their data through the classical survey technique supported by online services. Mostly, these latter ones are included in the "2. Health" cluster.

Of the studies that collected data from digital platforms (n = 172), 140 (81.39%) belong to cluster 1, "1. Information & Mass Media", making this branch of the discipline the one that most extensively employs innovative methodologies and draws on digital sources to conduct its research. The remaining 32 studies were quantitatively distributed across clusters 2, 4, 3, 6, and 5.

Figure 5 illustrates the use of the most frequent platforms from which data were extracted for the research. In 40% of the cases, data are collected from social media platforms such as YouTube, Twitter, Instagram, Facebook, and TikTok. Secondly, studies focusing on the analysis of video conferencing platforms account for nearly 17%, often involving in-depth interviews through the platform due to the impossibility of direct interaction. There are also some exceptional and experimental cases of ethnographies conducted through Zoom, the only mentioned video conferencing platform. The third

position is held by research extracting information from blogs, forums, and websites (16%), with a notable emphasis on the popular forum Reddit. Instant messaging takes the fourth spot, accounting for 9%, primarily involving applications like WhatsApp and WeChat, which is popular in China. In the fifth position are studies using digital newspapers as a source of analysis (11%), and finally, in smaller quantities, there are articles that used data from Google searches and digital commerce platforms like Amazon, accounting for 6% and 3% of cases, respectively.

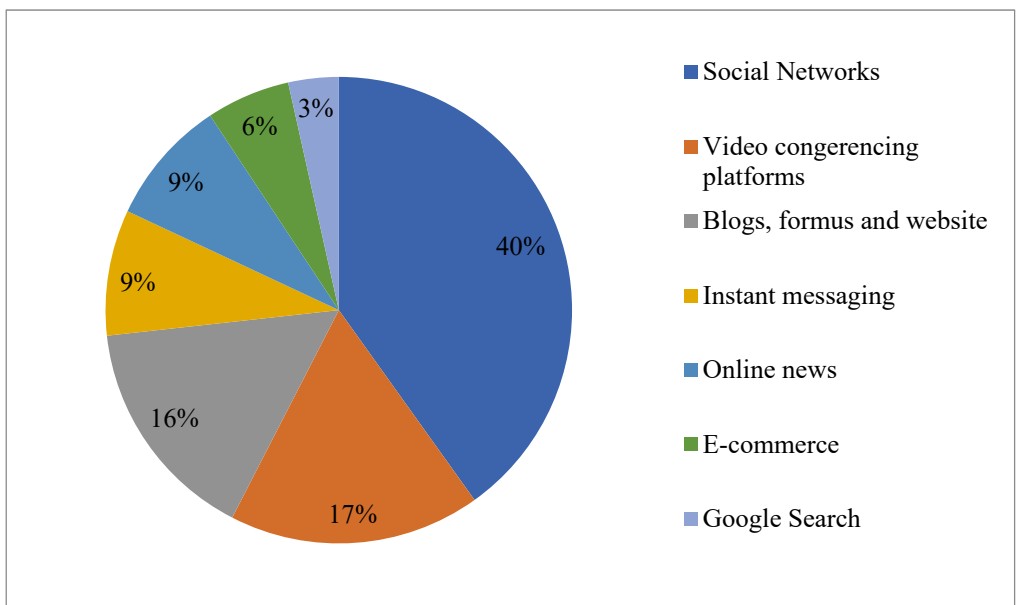

**Figure 5.** Main digital sources for data extraction.

Among the various platforms, it is notable that Twitter was the most frequently used, significantly more so than the rest. Figure 6 displays the top five digital applications used, with Twitter leading, followed by Facebook, Zoom, Instagram, and YouTube, in that order.

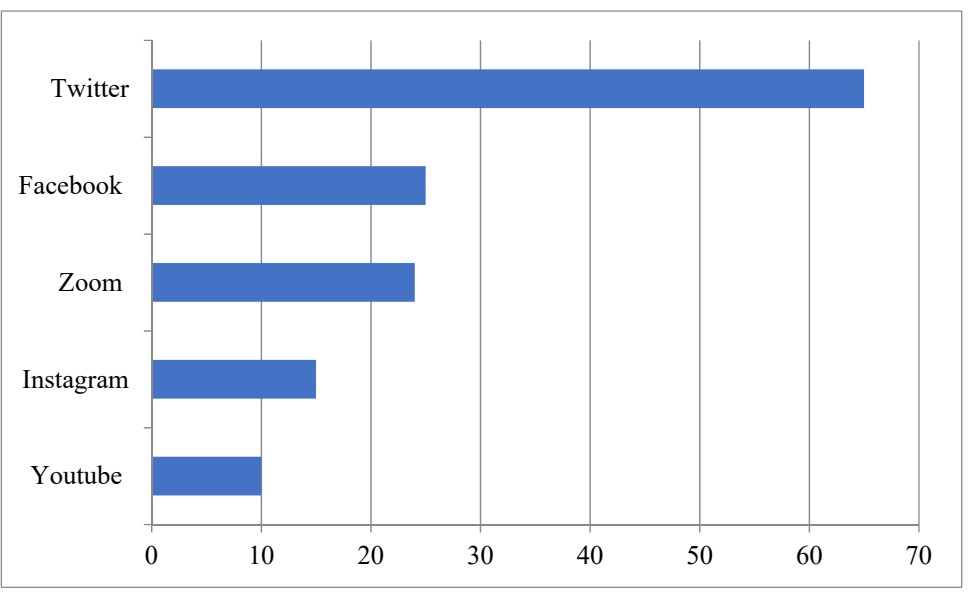

**Figure 6.** Top 5 digital platforms for data extraction.

From the analysis of the works that use social networks as a data source and the object of study, it is worth noting that even though the data were digital, most of the analyses

conducted fell within the classical methods of social sciences, with content analysis and a qualitative approach being prominent. Within a computational approach, the most employed methods were network analysis and topic modeling, although machine learning techniques were also applied for the prediction of tweets by categories. The rest of the articles were predominantly theoretical, focusing on the analysis of official and political discourses or reviews of previous works.

## 5. Discussion and Conclusions

This study is framed within a theoretical approach that advocates for the need to delve deeper into the conceptual frameworks of digital communication studies from a sociological perspective. In this context, it is argued that social sciences have forged a narrative that has permeated the collective imagination of societies regarding the understanding of the pandemic's impact on communicative and informational dynamics among social agents. This narrative has conceptualized the pandemic as a "digital revolution" (Wright 2020; Hantrais et al. 2021). However, this work maintains the premise that this revolution was already underway before the emergence of COVID-19, and that the pandemic merely highlighted the robustness of our interactions in the virtual environment (Torres-Albero 2022; Lobera and Torres-Albero 2021).

To elucidate this controversy and clarify the confusion surrounding certain data, it is crucial to recognize that the pandemic indeed marked a turning point in social science studies on digital communication. This is evident in both the field of political studies and in social networks. These research areas captured the attention of the social sciences due to the impossibility of conducting empirical research outside the virtual space and because much of the public debates with repercussions in the offline world were developed in that space.

To address this dispute, the present work is based on a systematic literature review of publications available in two databases of internationally renowned scientific journals, specifically those classified as sociological or related to this discipline, in relation to the study object of COVID-19.

The use of natural language processing and clustering techniques has enabled the identification of six major thematic groups that sociologists have highlighted as relevant to understanding the pandemic's impact. These themes have been of particular interest in sociological research on COVID-19.

Within the studies conducted, classic themes that have gained relevance in the context of the pandemic have been identified. These include aspects such as work, care, intra-family environments, education, and tourism. However, during the research on the pandemic, additional social dimensions have emerged that have acquired significant weight. Among these, the theme of health stands out, leading to extensive interdisciplinary work by sociologists.

In relation to our study hypothesis, it has been found that studies on information and media have led the understanding of the pandemic's impact This implies a heightened significance accorded by sociologists to communication studies, directly correlated with empirical observations of an increase in information consumption, even among individuals unaccustomed to them, during the pandemic (Casero-Ripolles 2020). Simultaneously, this underscores the relevance of these studies in comprehending social reality. The works included in the information and media cluster have approached various analytical paths, such as studies on political polarization, online hatred, fake news, and conspiratorial thinking (Zhang et al. 2021; Casino 2022).

These results may represent a significant advancement in the field of digital communication studies in the sociological domain. However, as mentioned in our theoretical framework, technological limitations still persist in the social sciences.

The analysis of data sources and methodologies used in empirical studies has revealed a limited use of digital data. Only 11.47% of the studies were based on digital data, and these were concentrated almost entirely in the information and mass media cluster, and computational analyses were even less frequent. This reflects a tendency toward more

traditional analysis strategies in sociology, which limits the scope of samples compared to the opportunities offered by new digital data sources.

It is important to clarify that this work cannot assert that the perception of a revolutionary change in digital life due to the pandemic has arisen solely due to increased research on these topics. Subsequent research should analyze to what extent changes in the use of digital platforms have been sustained after the pandemic and verify exogenous variables that truly address the correlation of a pandemic with a structural change in a digitally mediated lifestyle. However, it points to a dysfunction in the social sciences in terms of their capacity to understand the reality unfolding in the virtual environment.

From a methodological perspective, this study employed advanced natural language processing techniques to analyze an extensive corpus of academic texts. Although these techniques provide valuable information and enable comprehensive quantitative analysis, it is essential to recognize the inherent limitations of any computational approach (van Atteveldt and Peng 2021). These limitations include the need to interpret the context in which the topics are framed and the possibility of overlooking important qualitative nuances present in the analyzed texts.

Nevertheless, it is important to emphasize that this work contributes to the establishment of new guidelines in empirical research strategies in the field of sociology. It moves toward a closer approach to the study of digital communication and the recognition of a shift toward topics that are expected to gain significant relevance in the coming years.

**Author Contributions:** Conceptualization, A.T.-V. and C.T.-A.; Methodology, A.T.-V.; Software, A.T.-V.; Validation, A.T.-V. and C.T.-A.; Formal analysis, A.T.-V.; Investigation, A.T.-V.; Resources, A.T.-V.; Data curation, A.T.-V.; Writing—original draft, A.T.-V.; Writing—review & editing, A.T.-V.; Visualization, A.T.-V.; Supervision, C.T.-A.; Project administration, C.T.-A.; Funding acquisition, C.T.-A. All authors have read and agreed to the published version of the manuscript.

**Funding:** This work is part of the project "Trust, scientific systems and denialism. Social factors of vaccination in epidemic contexts. CONCERN" PID2020-115095RB-I00, funded by the Ministry of Science and Innovation and the State Research Agency/10.13039/501100011033/. It also has support PRE2021-097610, financed by MCIN/AEI/10.13039/501100011033 and the ESF+.

**Institutional Review Board Statement:** Not applicable.

**Informed Consent Statement:** Not applicable.

**Data Availability Statement:** Publicly available datasets were analyzed in this study. This data can be found here: https://github.com/AlbaTaboada/Digital-Communication-Studies-during-the-Pandemic (accessed on 1 December 2022).

**Conflicts of Interest:** The authors declare no conflicts of interest.

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
