# Peer review of "Digital Communication Studies during the Pandemic: A Sociological Review Using Topic Modeling Strategy"

_socsci, doi:10.3390/socsci13020078_

Round 1

Reviewer 1 Report

Comments and Suggestions for Authors

This paper presents some novel data that should be published with revision of approach and claims that are currently holding it back.

To start, I am uncomfortable labelling "oral process as the seminal spectrum of our communication.... and the digital social net works as the ultimate process resulting." Past the questionable grammar, the latter are in no way 'ultimate' and I'm not sure how oral processes of comms were seminal. This of course doesn't consider that what is at stake in the paper is technologically mediated comms; or what is termed 'virtual environments'? What is virtual about online, mass media on TV,  Newspapers, oral, or "soma"-based comms might better be explained?  Clarity would help the argument, and draw its frame from the assumptions held over from mid 1990s 'cyberspace' debates.

demoscopic needs to be defined at its first instance (46).

"For the first time, questions arose about who should be prioritized in accessing essential health resources and what their sociodemographic profiles should be (Núñez et 72 al. 2021; Pujolar et al. 2022)." 71

Is an overgeneralised claim (see Development Studies; debates on private-public health in developed nations). From Nunez' abstract the claim seems to be that "new barriers to access and the exacerbation of existing ones" were experienced vis-a-vis COVID-19 Crisis ; authors might wish to revise for accuracy over sensationalism.

claims of centralised instutions including"military ones, playing a key part in containing movement flows and consumption activities." needs to be cited/contextualised - did most of world, Rest of World, or the Free World have tanks in streets or MPs on crosswalks?.

claim of "In this context, four fundamental objectives intertwined in managing the health emergency" (90) should themselves contextualied; from where was this informed? And how does this relate to the overall two RQs, and why is LDA clustering useful (to think about whether the four objectives were researched??)

"The data indicating that the pandemic marked a turning point in general digital communications does not confirm the widespread and consolidated adoption of new technologies in everyday life. Instead, they reveal that these skills were already widely integrated into people's daily practices. Since it became impossible to continue life in a mixed format between digital and physical presence, the online digital space became the central element in human interactions."

Is a very well structured argument and might want to introduce the paper in a future revision - and again re 160-165. However, many frames/claims to the research within do not follow this argument;

Although the results are quite interesting, the authors' seem to view (information) technology seperate to life and this becomes a problem in the research design.

The "primary objective set forth in this study was to identify the main topics investigated from the sociological discipline regarding the health crisis caused by COVID-19", 289 and the methods to do so seem adequate and produce interesting (publishable) results.

The "The second aim of our study was to investigate the extent to which academic research related to information and media in the context of the pandemic focused on "digital" subjects of study. ...For this purpose, we selected articles grouped under cluster 1 "information and media," ". This inclusion/exclusion move presents methodological problems; health, education, and 'work/family/care' clusters from NLP would, considering the literature and expectations above, also contain content about media and information and/or digital subjects?

As an example authro states at 371 "reconciling telework forced by the health crisis with care in the family and domestic environment" in the health cluster. How are these subjects not digital vis-a-vis their telework? Or, in 382 re education "adapting classes and learning in a virtual environment during school closures" what part is not digital information or digitally mediated? 

In short, Information and Mass Media cluster might more be to do with "platform studies" re COVID? I guess another way to put is a 'digital subject' needs to be more clearly defined - are we talking cyborgs (Haraway)? or digital content? or digital platforms? or is it really a query on the "use of digital data" (493) in research and is this possible to know from your clusters? (ie. where digital methods were used or not doens't seem to be what the LDA did). Further, The keywords presented for #1 cluster have very little to do with those 'digital' things, from one view. Here authors' might want to consider work like ways of doing ethnographies *for* the internet (Hine 2015) to frame the subject of study vis-a-vis research on the pandemic? or take their advice that all kinds of "topics have gained great importance and have ceased to be exclusive to virtual spaces to manifest in offline life."

To that point, it might be pertinent to try to define offline life, and if authors are unable to do so in the context of 2020 pandemic, reframe their RQs.

Comments on the Quality of English Language

some phrases need revision but mostly fine.

Reviewer 2 Report

Comments and Suggestions for Authors

The research article presented is correct, well-structured and well-argued.

However, some suggestions for improvement are noted, such as the following:

At the content level

Title: the second part of the title is perceived as complex and difficult to understand considering that it is the title, which should make the research proposal clear from the beginning. It is recommended that the title be clarified, especially the second part as mentioned above. 

Abstract: is correct, as it synthesizes the most relevant information about the introduction, methodology, findings, and conclusions. It is recommended to include the main objective of the research in the body of the abstract.

Introduction:

-the introduction fulfils its function, as it situates the reader in relation to the object of study.

-it contextualizes the importance of communication in the field of sociology, and the role of information and communication technologies today.

-it emphasizes the challenge that the current communication landscape represents for the field of sociology.

-it highlights the period of the pandemic, an unprecedented situation that transformed communication and other facets of society. The fundamental role of communication in the prevention and containment of the crisis, with an emphasis on the digital.

-the relevance of the research presented is justified.

-the sources consulted are relevant and up to date, as they deal with a recent object of study.

Materials and methods:

-it is recommended to consider the inclusion of other previous research works that apply a similar methodology, so that the suitability of the same for the object of study is evidenced.

-the inclusion of sections explaining the work process that has been carried out at the methodological level is appreciated. The process is precisely detailed.

-the rigorousness of the process is acknowledged, which is laborious and complex, especially regarding the cleaning of the units of analysis.

-table 1 and figure 1 are adequate and facilitate understanding.

-table 2 lists the words or combinations that have been eliminated, according to a previous explanation in the text, but some of them still appear, such as "All right reserved", Taylor Francis Group, etc. It is not clear why text and table seem to contradict each other.

-the bibliographical sources that are referenced give authority and solvency to this methodology section.

Results:

-the results on the main sociological topics in relation to the pandemic, as well as the sub-themes, are interesting. 

-the work points out the characteristic of the transversality of some themes, as is then reflected in the topics. It is important that this is mentioned and anticipated in the paper.

-it is recommended that the percentage of the topics be repeated in each of the explanations, so as not to have to refer back to the table each time.

-as for the section on digital communication studies, figure 5 is redundant with the text on platforms. The text should incorporate some complementary information, not a simple repetition. There is a typo in the last figure of the results because it is figure 6 (not 5).

Discussion and conclusions:

-it is recommended in the introduction of this section to include some further reference to previous work, together with self-citations.

-the methodology used and its contribution to the study are discussed.

-the starting hypothesis is recovered and contrasted.

-highlight the significance of the main findings of the research.

-acknowledges limitations. It is recommended to incorporate possible future lines of research.

At the formal level

-some words are cut at the end of the sentence incorrectly, without respecting the syllables. A revision in this respect is recommended.

-in references, it is recommended that the way in which the doi of the papers are presented should be aligned.
